# Boosting Serotonin Synthesis Is Not Sufficient to Improve Motor Coordination of *Mecp2* Heterozygous Mouse Model of Rett Syndrome

**DOI:** 10.3390/biom14101230

**Published:** 2024-09-29

**Authors:** Claudia Villani, Giuseppina Sacchetti, Roberto W. Invernizzi

**Affiliations:** Istituto di Ricerche Farmacologiche Mario Negri IRCCS, Via Mario Negri, 2, 20156 Milan, Italy; claudia.villani@marionegri.it (C.V.);

**Keywords:** motor dysfunction, rare diseases, brain serotonin, tryptophan, α-lactalbumin

## Abstract

Motor deficit is a core symptom of Rett syndrome, a rare neurological disease caused in most cases by mutations of the *methyl-CpG-binding protein2* (*MECP2*) gene. Serotonin reuptake inhibitors improve motor coordination in *Mecp2* heterozygous (Het) mice and serotonin depletion prevented this effect. Here, we assess alterations in indole levels in various brain regions and whether boosting brain serotonin synthesis with the serotonin precursors tryptophan, 5-hydroxytryptophan and α-lactalbumin rescued motor coordination deficit of *Mecp2* Het mice. Motor coordination was assessed in the accelerated rotarod during and after systemic administration of serotonin precursors for 2–3 weeks. Since no data are available, the effect of α-lactalbumin on tryptophan, serotonin and 5-hydroxyindoleacetic acid levels was evaluated in various brain regions in order to identify the dose of ALAC to evaluate on motor coordination. As compared to WT, *Mecp2* Het mice show reduced levels of serotonin in the whole brain, hippocampus, brainstem and cerebral cortex, but not the striatum. Reduced levels of 5-hydroxyindoleacetic acid were observed in the hippocampus and brainstem. Doses of serotonin precursors increasing brain tryptophan and/or serotonin production and metabolism had no effect on motor coordination. The results indicate that boosting serotonin synthesis is not sufficient to improve motor coordination of *Mecp2* Het mice.

## 1. Introduction

Motor skill impairment is one of the core symptoms of Rett syndrome (RTT), a rare neurological disease caused in most cases by loss-of-function mutations of the methyl-CpG-binding protein 2 gene (*MECP2*; *Mecp2* in rodents) [1]. Motor symptoms observed in patients, including limited motility, ataxic gait, motor incoordination and tremors, are reproduced in most mouse models of the syndrome, with a good degree of face validity [2]. Low levels of serotonin (5-HT; 5-hydroxytryptamine), tryptophan hydroxylase 2 (*tph2*) expression and 5-hydroxyindoleacetic acid (5-HIAA), the main metabolite of 5-HT, are found in brain tissue and/or cerebrospinal fluid of RTT patients carrying *MECP2* mutations [3]. Consistently, alterations of brain levels, synthesis and release of 5-HT, and 5-HT receptor expression and function are observed in *Mecp2* null mouse models of RTT [4,5,6,7,8]. The limited information available in female *Mecp2* Het mice shows reduced tissue levels of 5-HT in the hippocampus but not in the raphe area [9], while no changes are found in brain 5-HT synthesis and cortical extracellular levels under basal condition or in response to a challenge dose of the selective 5-HT reuptake inhibitor (SSRI) fluoxetine [4].

Experimental studies in female *Mecp2* mutant mice show an improvement of motor, cognitive and respiratory symptoms in response to 5-HT receptor agonists and drugs enhancing the endogenous tone of 5-HT at brain synapses such the SSRIs and the mixed serotonergic and noradrenergic antidepressant mirtazapine [10,11,12]. Case reports support the potential use of serotonergic drugs including fluoxetine, 5-HT_1A_ receptor agonists or their combination for the treatment of respiratory deficits and stereotypies in patients with RTT [13,14,15] and SSRIs are considered the most effective drugs for the treatment of psychiatric symptoms of RTT [16].

We recently showed that repeated but not single administration of fluoxetine abolishes the motor coordination deficit in female *Mecp2* Het mice and enhances the expression of the MeCP2 protein in several regions of *Mecp2* Het mice [4,17]. These effects were no longer present after serotonin depletion with the 5-HT synthesis inhibitor p-chlorophenylalanine [4,17] suggesting that 5-HT plays a key role in rescuing motor coordination impairment. However, some of the effects of fluoxetine, such as its action on ionic channels and neurosteroid synthesis, are independent of endogenous 5-HT [18,19]. Thus, it is not yet clear whether the enhancement of brain 5-HT transmission is in itself sufficient to rescue motor coordination deficit of *Mecp2* Het mice or other factors may contribute.

Brain 5-HT synthesis depends, among other factors, on the brain concentration of the amino acid precursor tryptophan (TRP), an essential amino acid and physiological substrate of Tph2, the rate-limiting enzyme in the synthesis of brain 5-HT [20]. Unfortunately, the efficacy of 5-HT precursors in increasing brain 5-HT is limited by their short half-life in humans and rodents [21,22,23] and gastrointestinal adverse effects may occur when relatively high doses of TRP and 5-hydroxytryptophan (5-HTP) are used to overcome their short-lasting effect [24].

α-lactalbumin (ALAC), a whey protein rich in tryptophan [25], is well tolerated and safely used to supports adequate growth in infants [26,27]. Prolonged ingestion of ALAC increased the ratio of TRP/large neutral amino acids (LNAAs) in human plasma, which favoured the entry of TRP into the brain. In addition, ALAC increased cortical TRP levels and 5-HT synthesis and release in the rat hypothalamus [28,29,30,31]. ALAC (250–1000 mg/kg) was effective in several models of epilepsy in rats and mice, and counteracted depressive-like behaviour and intestinal inflammation in mice [30,31,32]. Thus, ALAC may be an alternative to TRP and 5-HTP to increase brain 5-HT synthesis/levels, with a safer profile of side effects.

The present study aims to provide further evidence on the alteration of 5-HT levels and metabolism and to evaluate the role of different 5-HT precursors in improving motor coordination in *Mecp2* Het mice. Since no data are available in the mouse, we assessed the effect of ALAC on brain indoles in order to establish the dose increasing brain TRP and 5-HT levels and metabolism to be used in a subsequent motor test in *Mecp2* Het mice.

## 2. Materials and Methods

### 2.1. Experimental Design and Statistics

Breeding pairs, consisting of female *Mecp2* Het (B6.129P2(C)-Mecp2tm1.1Bird/J; stock: 003890) and male C57BL/6J (stock: 000664) mice, were purchased from The Jackson Laboratory (Bar Harbor, ME, USA). Details of mice breeding and husbandry are described elsewhere [4]. All experimental data were obtained in female mice. As outlined in Figure 1, different cohorts of mice were used to identify alterations of brain monoamines and tryptophan in *Mecp2* Het mice as compared to WT, the effects of ALAC on brain monoamines and brain and plasma tryptophan and the ability of 5-HT precursors to counteract the motor coordination deficit of *Mecp2* Het mice.

Cohort 1 was used to assess differences in indole levels in the brain and selected brain areas of untreated WT and *Mecp2* Het mice. Cohort 1 was composed of 4 WT and 6 *Mecp2* Het mice aged 23–26 weeks. The dose- and time-dependent effects of a single oral administration of ALAC on brain and plasma levels of TRP and brain levels of indoles was assessed in 51 WT mice (Cohort 2a). Additional *Mecp2* Het mice (*n* = 10) were used to compare the efficacy of a single oral dose of ALAC (8000 mg/kg) in increasing plasma and brain levels of TRP and brain indole levels in WT (*n* = 12) and *Mecp2* Het mice (Cohort 2b). Cohort 2c consisted of 12 WT and 12 *Mecp2* Het mice aged 16–24 weeks given TRP once daily for 14 days and euthanised 1 h after the last dose of the chronic schedule to evaluate changes in brain TRP, 5-HT and 5-HIAA levels after chronic administration. The effect of TRP and 5-HTP on motor coordination was assessed in Cohort 3 mice, which included 34 wild type (WT) and 31 *Mecp2* Het mice aged 16–20 weeks. Mice were given TRP intraperitoneally once daily for 14 days and their performance on the rotarod was assessed 30–90 min after TRP administration. 5-HTP was administered in the drinking water, which was freely available during the 14 days of treatment. Rotarod performance of mice receiving TRP, 5-HTP or the respective vehicle was assessed 1, 3, 7 and 14 days after treatment.

The effect of ALAC on motor coordination was assessed in a cohort of mice consisting of 10 WT and 19 *Mecp2* Het mice aged 18–23 weeks (Cohort 4). ALAC was administered in the drinking water, freely available during the 21 days of treatment, and the latency to fall from the rotarod was assessed 1, 7, 14 and 21 days after treatment.

The effect of the various treatments on the rotarod (4–40 rpm in 300 s; Ugo Basile, Italy), was assessed in WT and *Mecp2* Het mice. On each testing day, mice underwent 4 trials interspersed by at least 15 min resting in the home cage between one trial and another. Mice were tested on the rotarod between 8:30 and 11:00 a.m. (in mice given TRP, the first trial started 30 min after injection). Figure 1 shows the schedule of treatment and the timing of behavioural assessments.

Mice were allocated to experimental groups by age-stratified randomization. The experimenter assessing rotarod performance was not blind to treatments.

Statistical analysis was performed using GraphPad Prism version 10.1.2 (GraphPad Software, Boston, MA, USA) or Stat-View 5.0 (SAS Institute Inc., Cary, NC, USA). One-way ANOVA followed by Dunnett’s test was used to compare the dose- and time-dependent effect of ALAC on TRP, 5-HT and 5-HIAA in WT mice. The effect of 8000 mg/kg ALAC on indole and TRP levels in *Mecp2* Het in comparison to WT mice were analysed by two-way ANOVA with genotype and treatment as between-subject factors, followed by Sidak’s test. Indole levels in untreated WT and *Mecp2* Het mice were compared by Student’s *t*-test.

The mean latency to fall of four daily trials was calculated for each mouse and compared across the experimental groups by 3-way ANOVA or a mixed-effects model, with days of treatment as a within-subject factor and genotype and treatment as between-subject factors. If ANOVA revealed a significant effect of treatment or an interaction treatment × genotype or treatment × time, further analyses were carried out to compare the effect of each treatment versus vehicle with Sidak’s test. Raw data are available in Zenodo (https://zenodo.org/uploads/13134000).

### 2.2. Drug Treatment

L-tryptophan (Fluka, Buchs, Switzerland) was dissolved in a minimal volume of 1 M NaOH and diluted with sterile, pyrogen-free water. The pH of the solution was adjusted to 6.5–7.0 with 1 M HCl. Appropriate volumes of concentrated (10×) phosphate buffered saline (Lonza, Belgium) was added and the solution was brought to final volume with pyrogen-free water. Mice received 300 mg/kg tryptophan or an equivalent volume (20 mL/kg) of vehicle intraperitoneally (i.p.) between 8:00 and 10:00 a.m., during the light phase of the light-dark cycle (light on at 7:00 a.m.). 5-HTP (Alfasigma, Italy) and ALAC (Dermolife, Italy) were dissolved and administered in drinking water. As 5-HTP solution is light-sensitive, drinking bottles were protected from light by covering them with aluminium foil and replaced daily. Bottles filled with ALAC solution were replaced with clean bottles containing fresh ALAC solution every 2–3 days. Mice had free access to bottles containing 5-HTP, ALAC or plain water 24 h a day. The mean fluid intake per mouse, expressed as the total volume of fluid consumed per cage (*n* = 2–4 mice/cage) and per day, was monitored by weighing the bottles every day (5-HTP) or every 2–3 days (ALAC). If mean fluid intake varied by more than 20%, the concentration of ALAC or 5-HTP was adjusted to keep constant the daily drug intake. The doses of TRP (300 mg/kg/day) and 5-HTP (100 mg/kg/day) used in the present study were previously shown to enhance brain 5-HT levels, synthesis and/or release in WT mice and in mice with reduced Tph2 activity [24,33].

### 2.3. Monoamines and Tryptophan Assay

Mice were euthanised by decapitation under deep anaesthesia (ketamine/medetomidine 75/1, mg/kg i.p.) between 9:00 and 12:00 a.m. (with the exception of the mice used to evaluate the effect of ALAC 8 h after administration, which were euthanised at 4 p.m.). Ketamine HCl (Lobotor) and medetomidine HCl (Domitor) were purchased from ACME SRL, Italy and Vetoquinol, Italy, respectively. The brain was rapidly removed from the skull and positioned on a refrigerated aluminium plate. The cerebellum was removed and the brain was divided into two hemispheres with a cut along the sagittal line. Half the brain was immediately frozen on dry ice. The cerebral cortex, hippocampus, striatum and brainstem were dissected freehand from the remaining hemisphere and frozen on dry ice. Trunk blood (0.5–1.0 mL) was collected in 1.5 mL tubes containing 5 mg NaF and 7 mg Na_2_EDTA·2H_2_O (Milian, Switzerland) to prevent clotting and stored at 4 °C for 30 min. Plasma was separated by centrifugation at 5000× *g* for 10 min, at 4 °C with an Eppendorf 5417R centrifuge (Eppendorf, Milan, Italy) and stored at −70 °C until analysis. Brain tissues were weighed, transferred to Eppendorf 3810 tubes and stored at −70 °C until analysis. Brain tissues were homogenized in 10 volumes (20 volumes for striatum and hippocampus) 0.1 M HClO_4_ (Carlo Erba, Milan, Italy) containing 0.1% Na_2_EDTA·2H_2_O (Sigma-Aldrich, Milan, Italy), 0.05% Na_2_S_2_O_5_ (Carlo Erba, Milan, Italy) and 0.002% ascorbic acid (Sigma-Aldrich, Milan, Italy) by an ultrasonic homogeniser (Sonoplus HD2070, Bandelin, Berlin, Germany). Homogenates were centrifuged (12,000× *g* for 10 min at 4 °C) with an Avanti J-E centrifuge (Beckman-Coulter, Cassina de’ Pecchi, Italy). The supernatant was filtered through 0.45 μm PVDF syringe filters (Perkin Elmer, Milan, Italy) and split into 2 aliquots for the determination of indoles and TRP, by reverse-phase high performance liquid chromatography coupled with electrochemical detection. Briefly, indoles were separated by a reverse phase column (Kinetex^®^ 2.6 µm C18 100 Å, LC Column 75 × 3 mm, Phenomenex, Bologna, Italy). The mobile phase consisted of 3.0 g/L sodium acetate·3H_2_O (Merck, Darmstadt, Germany), 5 g/L citric acid monohydrate (Merck, Darmstadt, Germany), 37 mg/L Na_2_EDTA·2H_2_O (Sigma-Aldrich, Italy), 250 mg/L sodium heptane sulfonate (Sigma-Aldrich, Milan, Italy). The pH of the solution was adjusted to 3.5 before the addition of 8% CH_3_OH (Merck, Darmstadt, Germany). The flow rate of the mobile phase was maintained at 0.4 mL/min with a LC-20AD isocratic pump (Shimadzu, Milan, Italy). The electrochemical detector (Coulochem II; ESA, Chemsford, MA, USA) was equipped with a 5011 analytical cell (E_1_ = 0 mV; E_2_ = +300 mV). Total levels of TRP in the half brain, brain regions and plasma were determined as previously described [34].

## 3. Results

### 3.1. Brain Levels of Monoamines and Tryptophan in Female WT and Mecp2 Het Mice

5-HT, 5-HIAA and TRP levels in the whole brain and various brain regions were compared in WT and *Mecp2* Het mice (Table 1).

*Mecp2* Het mice aged 23–26 weeks at the time of sacrifice show reduced levels of brain 5-HT, but no significant changes in brain 5-HIAA and TRP in the whole brain. Regional analysis of monoamine levels showed a reduction of about 20% of 5-HT in the hippocampus, cerebral cortex and brainstem, while no significant changes were observed in the striatum. 5-HIAA levels were reduced in the hippocampus and brainstem by 20% and 11%, respectively, but no reduction was observed in the cerebral cortex and striatum.

**Table 1 biomolecules-14-01230-t001:** Levels of 5-HT, 5-HIAA and TRP in the whole brain and brain regions of WT and *Mecp2* Het mice.

		5-HT	5-HIAA	TRP
		ng/g	ng/g	μg/g
**BRAIN**				
WT (4)		839 ± 26	456 ± 26	4.5 ± 0.2
HET (6)		**679 ± 21**	391 ± 24	4.5 ± 0.3
	*t(df)*	*4.763(8)*	*1.803(8)*	*0.0795(8)*
	*p*	*0.0014*	*0.109*	*0.938*
**HIPP**				
WT (4)		792 ± 43	505 ± 21	5.2 ± 0.3
HET (6)		**617 ± 49**	**407 ± 18**	4.9 ± 0.2
	*t(df)*	*2.484(8)*	*3.521(8)*	*0.8481(8)*
	*p*	*0.0379*	*0.0078*	*0.8481*
**CTX**				
WT (4)		709 ± 19	285 ± 8	3.3 ± 0.1
HET (6)		**580 ± 26**	263 ± 13	3.6 ± 0.2
	*t(df)*	*3.607(8)*	*1.284(8)*	*1.169(8)*
	*p*	*0.0069*	*0.2351*	*0.2761*
**BST**				
WT (4)		1063 ± 37	556 ± 24	3.8 ± 0.1
HET (6)		**817 ± 34**	**493 ± 13**	4.2 ± 0.2
	*t(df)*	*4.778(8)*	*2.549(8)*	*1.290(8)*
	*p*	*0.014*	*0.0342*	*0.2332*
**STR**				
WT (4)		674 ± 104	403 ± 55	5.0 ± 0.1
HET (5)		601 ± 68	364 ± 24	5.0 ± 0.4
	*t(df)*	*0.7250(7)*	*0.6720(7)*	*0.0668(7)*
	*p*	*0.492*	*0.5231*	*0.9486*

Data are mean ± SEM. The number of mice/group is shown in parentheses. 5-HIAA, 5-hydroxyindoleacetic acid; 5-HT, 5-hydroxytryptamine; BST, brainstem; CTX, cerebral cortex; HET, *Mecp2* Het; HIPP, hippocampus; STR, striatum; TRP, tryptophan WT, wild type. Bold type indicates a significant difference vs. WT (Student’s *t*-test). T-values, degrees of freedom (df) and associated probability (*p*) are indicated in italics.

### 3.2. Dose-Dependent Effect of Single Oral Doses of ALAC on Plasma and Brain Levels of TRP and Brain Levels of 5-HT and 5-HIAA in WT Mice

To evaluate the effect of ALAC on brain indoles, WT mice were treated with single doses of 250–8000 mg/kg per os. The dose of ALAC increasing brain TRP and indoles in WT mice (8000 mg/kg) was subsequently evaluated in *Mecp2* Het mice to confirm its efficacy in mutant mice.

The dose-dependent effect of ALAC on plasma and brain TRP levels and brain 5-HT and 5-HIAA levels in WT mice is shown in Figure 2. A single administration of 8000 mg/kg ALAC significantly increased TRP levels in the plasma (F3,16 = 30.05; *p* < 0.0001; one-way ANOVA) and brain (F3,15 = 13.57; *p* = 0.0002; one-way ANOVA) of WT mice 2 h after the administration, reaching 323% and 279% of control levels, respectively. Lower doses had no significant effect. At 8000 mg/kg, ALAC significantly increased brain levels of 5-HT (115% of control levels; F3,16 = 9.032; *p* = 0.001; one-way ANOVA) and 5-HIAA (142% of control levels; F3,15 = 20.66; *p* < 0.0001; one-way ANOVA) in WT mice. Lower doses of ALAC had no significant effect on brain 5-HT and 5-HIAA.

**Figure 2 biomolecules-14-01230-f002:**
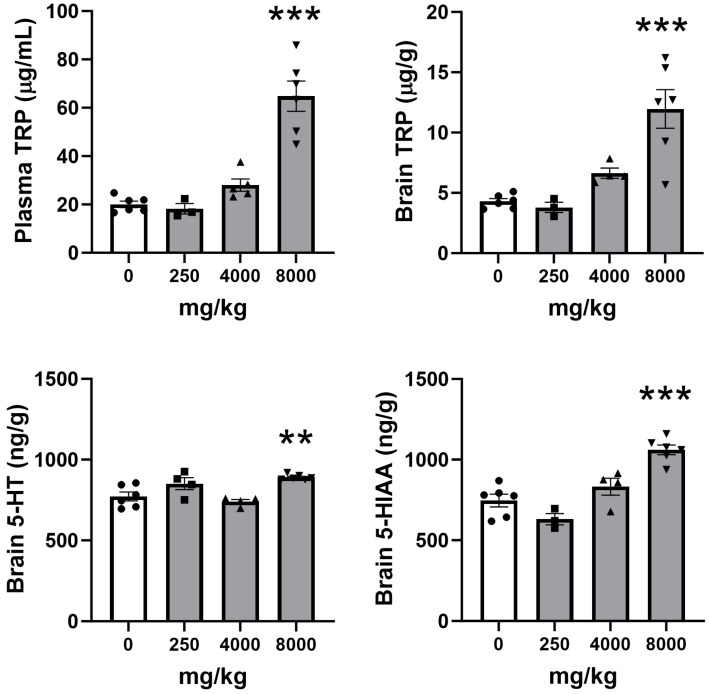
Dose-dependent effect of single oral doses of α-lactalbumin (ALAC; 250–8000 mg/kg) on plasma and brain tryptophan (TRP) and brain 5-hydroxytryptamine (5-HT) and 5-hydroxyindoleacetic acid (5-HIAA) levels in wild type mice. Data are mean ± SEM of 3–6 mice/group. Solid circles, squares and triangles represent individual data. F-values (one-way ANOVA) are shown in Results. ** *p* = 0.0095; *** *p* < 0.0001 vs. vehicle (0 mg/kg), Dunnett’s test.

### 3.3. Time-Dependent Effect of ALAC on Plasma and Brain Levels of TRP and Brain Levels of 5-HT and 5-HIAA in WT Mice

Figure 3 shows the time-dependent effect of a single oral dose of ALAC (8000 mg/kg) on plasma and brain levels of TRP and brain levels of 5-HT and 5-HIAA in WT mice. ALAC significantly increased TRP in plasma (F4,24, 90.39; *p* < 0.0001) and brain (F4,24, 23.98; *p* < 0.0001) and brain 5-HT (F4,24, 7.158; *p* = 0.0006) 1 and 2 h after administration. A significant increase in 5-HIAA (F4,24, 19.99; *p* < 0.0001) was observed only 2 h after ALAC administration. Plasma TRP levels were maximally increased 1 h after dosing, while brain TRP, 5-HT and 5-HIAA increased to a similar extent 1 or 2 h after administration. ALAC had no effect on TRP, 5-HT and 5-HIAA levels 4 and 8 h after dosing.

**Figure 3 biomolecules-14-01230-f003:**
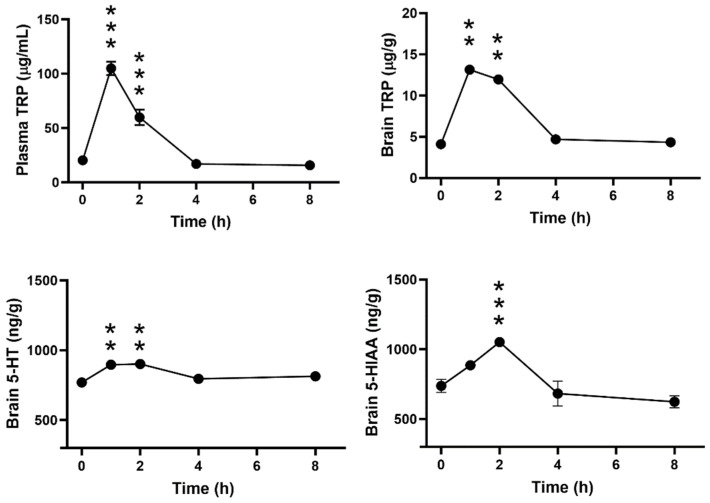
Time-dependent effect of a single oral dose of α-lactalbumin (ALAC; 8000 mg/kg) on plasma and brain levels of tryptophan (TRP) and brain levels of 5-hydroxytryptamine (5-HT) and 5-hydroxyindoleacetic acid (5-HIAA) in wild type mice. F-values (one-way ANOVA) are shown in Results. Data are mean ± SEM of 5–7 mice/group. ** *p* < 0.002; *** *p* < 0.0001 (Dunnett’s test).

### 3.4. Effect of 8000 mg/kg ALAC on Plasma and Brain Levels of TRP and Brain Levels of 5-HT and 5-HIAA in WT and Mecp2 Het Mice

Figure 4 shows the effect of a single oral dose of ALAC (8000 mg/kg) on plasma and brain levels of TRP and brain levels of 5-HT and 5-HIAA in WT and *Mecp2* Het mice. ALAC significantly increased plasma and brain levels of TRP in WT mice reaching 323% and 279% of control mice, respectively. ALAC had a similar effect in *Mecp2* Het mice where the increase in TRP reached 349% and 313% of control values in plasma and brain, respectively. Two-way ANOVA showed a significant effect of treatment on plasma (F1,18 = 118.8, *p* < 0.0001) and brain (F1,18 = 43.43, *p* < 0.0001) TRP, but no significant effects of genotype (Plasma, F1,18= 1.582, *p* = 0.2246; Brain, F1,18 = 1.709, *p* = 0.2076) and genotype × treatment interaction (Plasma, F1,18 = 0.1621, *p* = 0.6920; Brain, F1,18 = 0.2584, *p* = 0.6174). ALAC significantly increased brain levels of 5-HIAA in WT mice (142% of control values) and in *Mecp2* Het mice (129% of control values). Two-way ANOVA showed a significant effect of treatment (F1,18 = 26.46, *p* < 0.0001), genotype (F1,18 = 22.69, *p* = 0.0002) but not genotype × treatment interaction (F1,18 = 2.323, *p* = 0.1449). ALAC significantly increased brain levels of 5-HT in WT (115% of control values) but had no significant effect in *Mecp2* Het mice. Two-way ANOVA showed a significant effect of treatment (F1,18 = 10.67, *p* = 0.0043), genotype (F1,18 = 22.57, *p* = 0.0002) but not genotype × treatment interaction (F1,18 = 2.213, *p* = 0.1542).

**Figure 4 biomolecules-14-01230-f004:**
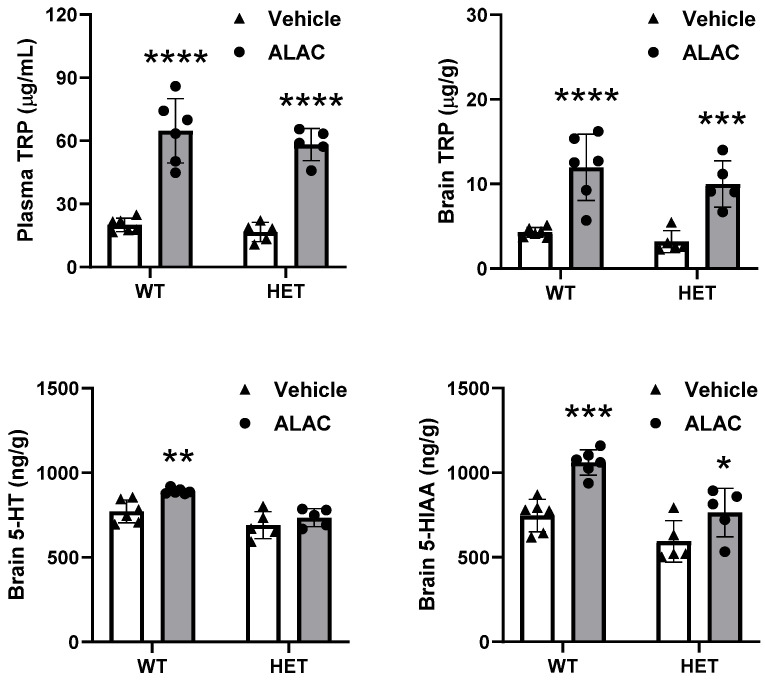
Effect of a single oral dose of α-lactalbumin (ALAC; 8000 mg/kg) on plasma and brain levels of tryptophan (TRP) and brain levels of serotonin (5-HT) and 5-hydroxyindoleacetic acid (5-HIAA) in wild type (WT) and *Mecp2* Het (HET) mice. Data are mean ± SEM of 5–6 mice/group. Solid circles and triangles represent individual data. F-values (two-way ANOVA) are shown in Results. * *p* < 0.05; ** *p* < 0.01; *** *p* < 0.001; **** *p* < 0.0001 vs. vehicle (Sidak’s test).

### 3.5. Effect of Repeated Treatment with Tryptophan on Brain Levels of TRP, 5-HT and 5-HIAA in WT and Mecp2 Het Mice

Table 2 shows the effect of TRP given i.p. once daily for 14 days on brain levels of TRP, 5-HT and 5-HIAA. Administration of exogenous TRP increased brain levels of TRP by 2858% and 2466% of control values in WT and *Mecp2* Het mice, respectively. Brain levels of 5-HT and 5-HIAA significantly increased in response to repeated administration of TRP reaching 129% and 240% of control values in WT and 139% and 219% in *Mecp2* Het mice. Two-way ANOVA showed a significant effect of treatment for TRP, 5-HT and 5-HIAA, a significant effect of genotype for 5-HT, but no significant effect of genotype x treatment interaction for TRP and indoles.

**Table 2 biomolecules-14-01230-t002:** Brain levels of **TRP**, **5-HT** and **5-HIAA** in WT and *Mecp2* Het mice receiving repeated doses of TRP for 14 days.

	TRP	5-HT	5-HIAA
	μg/g	ng/g tissue	ng/g tissue
WT-Veh (5)	3.4 ± 0.4	601 ± 31	668 ± 74
WT-TRP (7)	97.2 ± 6.3 ***	773 ± 29 **	1602 ± 114 ***
HET-Veh (6)	3.5 ± 0.2	631 ± 18	613 ± 31
HET-TRP (6)	86.3 ± 8.2 ***	876 ± 41 ***	1343 ± 89 ***
*F treatment (df), p*	*248.4 (1,20), p < 0.0001*	*44.33 (1,20), p < 0.0001*	*89.41 (1,20), p < 0.0001*
*F genotype (df), p*	*0.9199 (1,20), p = 0.3490*	*4.543 (1,20), p = 0.0457*	*3.176 (1,20), p = 0.0899*
*F interaction (df), p*	*0.9581 (1,20), p = 0.3394*	*1.406 (1,20), p = 0.2497*	*1.330 (1,20), p = 0.2623*

Mice received 300 mg/kg tryptophan intraperitoneally once daily for 14 days and were euthanised 1h after the last dose of the chronic schedule. Data are mean ± SEM. The number of mice/group is shown in parentheses. 5-HIAA, 5-hydroxyindoleacetic acid; 5-HT, 5-hydroxytryptamine; HET, *Mecp2* Het; TRP, tryptophan; Veh, vehicle; WT, wild type. F values (two-way ANOVA), degrees of freedom (df) and associated probability (*p*) are indicated in italics. ** *p* = 0.005, *** *p* < 0.0001 vs. respective vehicle (Tukey’s multiple comparisons test).

### 3.6. Effect of TRP, 5-HTP and ALAC on Rotarod Performance in Mecp2 Het Mice

The effect of TRP and 5-HTP on the latency to fall from the accelerated rotarod was examined in WT and *Mecp2* Het mice at various time points after administration (Figure 5). As previously reported [4], the performance of untreated *Mecp2* Het mice in the rotarod is much lower than in WT mice (mean latency to fall: *Mecp2* Het, 159 ± 7 and WT, 273 ± 4 s). Consistently, ANOVA showed a highly significant effect of genotype (F1,59 = 98.143, *p* < 0.0001).

TRP and 5-HTP had no significant effect on the latency to fall of WT and *Mecp2* Het mice at any times after treatment (Figure 5). Three-way ANOVA showed a significant effect of days (F4,236 = *p* = 0.0037), but no effects of treatment (F2,59 = 1.591, *p* = 0.2123), genotype × treatment (F2,59 = 1.113, *p* = 0.3354), days × genotype (F4,236 = 2.328, *p* = 0.0569), days × treatment (F8,236 = 0.549, *p* = 0.8189) and days × genotype × treatment (F8,236 = 0.688, *p* = 0.7019) interaction.

**Figure 5 biomolecules-14-01230-f005:**
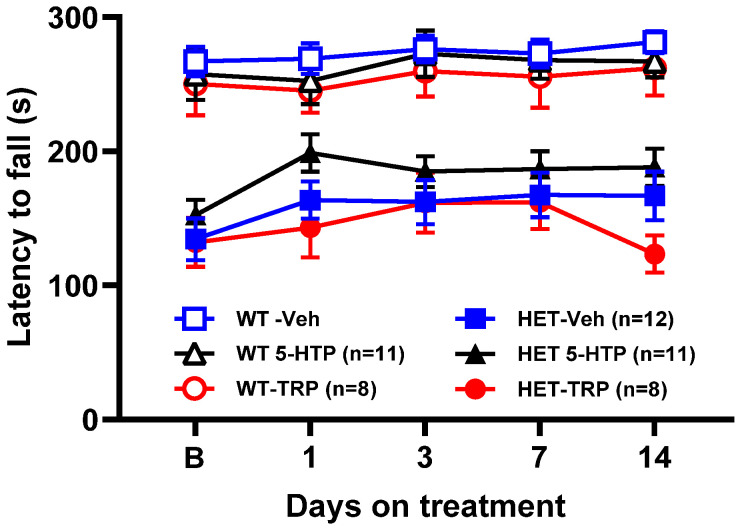
Effect of tryptophan (TRP; 300 mg/kg, once daily) and 5-hydroxytryptophan (5-HTP; 100 mg/kg/day in drinking water) on the latency to fall from the accelerated rotarod in wild type (WT) and *Mecp2* Het (HET) mice. F-values (three-way ANOVA) are shown in Results. Each time-point is the mean ± SEM of four daily trials. The number of mice per group is shown in parentheses. B (basal), latency to fall before the start of treatment; Veh, vehicle.

ALAC had no effect on the latency to fall of WT and *Mecp2* Het mice at any times after administration (Figure 6). Three-way ANOVA showed a significant effect of genotype (F1,25 = 13.277, *p* = 0.0012), but not treatment (F1,25 = 0.022, *p* = 0.8844), days (F4,100 = 1.047, *p* = 3871), genotype × treatment (1,25 = 0.002. *p* = 0.9675), days × genotype (F4,100 = 0.768, *p* = 0.5487), days × treatment (F4,100 = 0.154, *p* = 0.9608) and days × genotype × treatment (F4,100 = 0.053, *p* = 0.9946) interaction.

**Figure 6 biomolecules-14-01230-f006:**
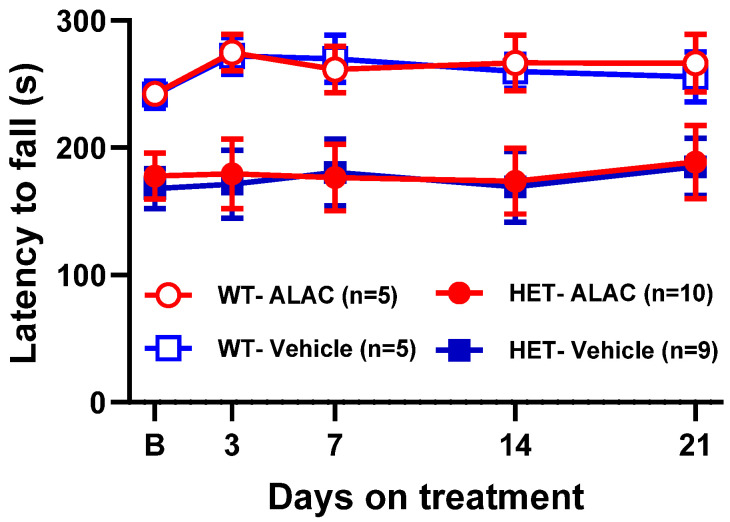
Effect of α-lactalbumin (ALAC; 8000 mg/kg/day in drinking water) in the accelerated rotarod in wild type (WT) and *Mecp2* Het (HET) mice. F-values (three-way ANOVA) are shown in Results. Each time-point is the mean ± SEM of four daily trials. The number of mice per group is shown in parentheses. B (basal), latency to fall before the start of treatment.

## 4. Discussion

This study demonstrates that 5-HT and 5-HIAA levels are reduced in adult *Mecp2* Het mice (5–6 months of age) and shows that treatment with 5-HT precursors does not modify the motor coordination deficit of *Mecp2* Het mice.

Although the reduction in 5-HT and 5-HIAA levels in the hippocampus of *Mecp2* Het mice confirm previous findings [9], to our knowledge no other studies assessed whether alterations of 5-HT homeostasis occur in the brain of female *Mecp2* Het mice. As compared to WT, *Mecp2* Het mice show lower levels of hippocampal, brainstem and cortical 5-HT. This indicates that 5-HT deficiency, albeit small, is widespread across different brain regions except the striatum where no differences are found between genotypes. The reduction in 5-HT levels and metabolism is small and likely not sufficient to account for the motor coordination deficit. Consistently, genetically and pharmacologically induced depletion of brain 5-HT has no effect on rotarod performance in WT mice and does not affect the motor coordination impairment of *Mecp2* Het mice [4,35]. As in previous studies in male *Mecp2* null mice [6,7], we found no changes in 5-HT and 5-HIAA in the striatum of *Mecp2* Het mice.

Several hypotheses have been advanced on the causes of brain 5-HT deficit observed in RTT patients and mouse models of the disease. These include reduced 5-HT innervation, production, reuptake and metabolism of 5-HT [6]. Although the role of these factors was not systematically addressed in the present study, data presented here and in other published papers may help clarify the role of some of these factors. We found no changes in TRP levels in any brain region investigated. Therefore, the reduced 5-HT levels are not caused by deficiency of the endogenous precursor. Reduced *tph2* gene expression has been found in male mice globally lacking the *Mecp2* gene and in mice with selective deletion of the gene in 5-HT neurons [3]. However, another study found no reduction in *tph2* expression in raphe nuclei of *Mecp2* null mice [6] and data on *tph2* expression in female heterozygous mice are lacking. We measured the levels of 5-HTP after l-aromatic amino acid decarboxylase inhibition, an indicator of 5-HT synthesis [36], and found no differences between WT and *Mecp2* Het mice in 5-HTP accumulation in the hippocampus, striatum, brainstem, prefrontal cortex and the rest of the cerebral cortex [4]. Thus, the low levels and metabolism of 5-HT found in most brain regions of adult *Mecp2* Het mice unlikely reflects reduced 5-HT synthesis.

No differences in cortical 5-HT innervation, synaptic markers and expression of 5-HT-related genes such as the serotonin transporter and monoamine oxidase, which may have marked effects on 5-HT storage and metabolism, have been observed in *Mecp2* null mice and no data are available in *Mecp2* Het mice [6,8]. Thus, the factors contributing to the reduction in indole levels observed in *Mecp2* Het mice remain largely unknown.

5-HT plays a key role in fluoxetine’s ability to rescue motor coordination impairment of *Mecp2* Het mice [4]. Based on this finding, we expected that 5-HT precursors may mimic the ability of fluoxetine to rescue motor coordination deficit in *Mecp2* Het mice but the data presented here do not support this hypothesis.

It is unlikely that the lack of effect of 5-HT precursors is due to underdosing as previous studies clearly showed that doses of TRP and 5-HTP similar to those used in the present study or lower increase the production and release of brain serotonin. In addition, TRP and 5-HTP restored normal 5-HT levels in mice with reduced function or total deletion of the *Tph2* gene [24,33,37,38,39,40,41].

Here, we provide evidence that a single dose of 8000 mg/kg ALAC increased brain levels of TRP in *Mecp2* Het mice by more than 3-fold. Brain 5-HT levels are only marginally increased by ALAC, while a larger effect is found on brain 5-HIAA, which indicates that ALAC stimulates the synthesis and metabolism of 5-HT. However, these effects are much lower than those achieved after the administration of 300 mg/kg TRP once daily for 14 days under the same experimental conditions used to assess the effect of TRP on motor coordination. Since repeated dosing of TRP had no effect on motor coordination, it is reasonable to conclude that boosting 5-HT synthesis is not sufficient to counteract the motor coordination deficit of *Mecp2* Het mice. Consistently, rats ingesting diets enriched in ALAC (roughly 4–30 g/kg/day) show increased brain TRP and 5-HT levels, synthesis and release of 5-HT [29,42]. According to dose translation from animal to human studies using the body surface area normalization method [43], 8000 mg/kg ALAC as used in this study is comparable to the dose increasing TRP in human plasma both in terms of effectiveness and duration [44]. To our knowledge, this is the first study providing information on dose- and time-dependent effect of ALAC on mouse brain and plasma TRP and brain 5-HT and 5-HIAA levels. Data clearly show that doses of ALAC lower than 8000 mg/kg do not affect brain and plasma TRP and brain levels and metabolism of 5-HT. This suggests that the effects of lower doses of ALAC as used in previous mouse studies can hardly be attributed to changes in TRP and 5-HT availability in the brain and peripheral organs.

5-HT plays an important role in early neuronal development and function including neuronal proliferation, migration and differentiation as well as motor skills [45,46,47]. Thus, it cannot be excluded that the administration of 5-HT precursors during the neonatal period or early childhood may be more effective in improving motor coordination in *Mecp2* Het mice compared to administration in adults.

Motor coordination improvement in *Mecp2* Het mice takes about a week to develop after fluoxetine or citalopram [4]. This suggests that adaptive changes developing over time likely underlie the ability of SSRIs to rescue motor coordination deficit. Consistently, changes in neuronal excitability, protein synthesis, plasticity or neurogenesis have been involved in the mechanisms facilitating the effect of 5-HT and SSRIs on motor learning [48] and are defective in patients with RTT and mouse models of the pathology [10,49]. The long half-life of fluoxetine and its active metabolite (6 and 12 h, respectively [50]) ensures prolonged exposure to the drug during the course of treatment. The half-lives of TRP and 5-HTP are short and their effects in rodents generally last a few hours or less [23,24,40,51,52]. Thus, it cannot be excluded that the failure of 5-HT precursors to improve motor coordination reflects their short-lived effect on brain 5-HT. However, the effectiveness of citalopram in improving motor coordination of *Mecp2* Het mice despite its short half-life (about 1.5 h [53]) argues against this interpretation. This suggests that factors other than duration of action could account for the different ability of 5-HT precursors and SSRIs to improve motor coordination in *Mecp2* Het mice.

Restoration of *Mecp2* expression globally or selectively in the striatum is sufficient to rescue rotarod impairment in *Mecp2* Het mice [54]. This suggests that the improvement in motor coordination produced by SSRIs likely reflects their ability to promote MeCP2 expression in several brain areas of the *Mecp2* Het mouse, including the striatum [4,17]. If this effect is not shared by 5-HT precursors, it could explain their ineffectiveness on motor coordination. Further studies are needed to confirm this hypothesis.

In summary, current data show that 5-HT precursors alone are not effective in improving motor coordination of *Mecp2* Het mice. TRP has no effect alone, but potentiates the antidepressant-like effect of SSRIs [33,55,56]. Therefore, it would be important to explore the possibility that, although ineffective alone, 5-HT precursors may improve the effect of SSRIs on motor coordination in RTT mice.

## Figures and Tables

**Figure 1 biomolecules-14-01230-f001:**
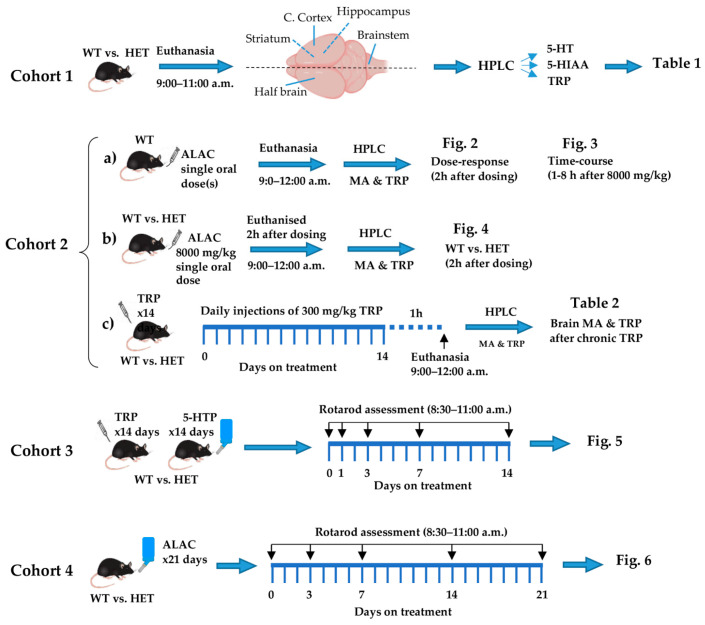
Schedule of treatment and timing of biochemical and behavioural assessments. Cohort 1 consisted of untreated wild type and *Mecp2* Het mice used to assess differences in regional brain levels of 5-HT, 5-HIAA and TRP between genotypes (Table 1). Cohort 2a was used to assess the dose- and time-dependent effect of ALAC on brain levels of 5-HT, 5-HIAA and brain and plasma levels of TRP in WT mice (Figure 2 and Figure 3). Cohort 2b was used to determine whether ALAC (8000 mg/kg) had a similar effect on monoamines and TRP in WT and *Mecp2* Het mice (Figure 4). Cohort 2c was used to assess the effect of a repeated treatment with TRP on brain indole and TRP levels (Table 2). Cohorts 3 and 4 were used to assess the effect of TRP (300 mg/kg i.p., once daily for 14 days), 5-HTP (100 mg/kg/day in drinking water) and ALAC (8000 mg/kg/day in drinking water) on motor coordination in WT and *Mecp2* Het mice (Figure 5 and Figure 6). The black arrows indicate days from the start of treatment when rotarod performance was measured. 5-HIAA, 5-hydroxyindoleacetic acid; 5-HT, 5-hydroxytryptamine; 5-HTP, 5-hydroxytryptophan; ALAC, α-lactalbumin; HET, *Mecp2* heterozygous; HPLC, high performance liquid chromatography; MA, monoamines; TRP, tryptophan; WT, wild type.

## Data Availability

The original contributions presented in the study are included in the article; further inquiries can be directed to the corresponding author/s. The raw data can be downloaded at: https://zenodo.org/uploads/13134000.

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
