# Peer review of "Boosting Serotonin Synthesis Is Not Sufficient to Improve Motor Coordination of Mecp2 Heterozygous Mouse Model of Rett Syndrome"

_biomolecules, 2024, doi:10.3390/biom14101230_

Round 1

Reviewer 1 Report

Comments and Suggestions for Authors

While the present manuscript has the potential to greatly contribute to the field, concerns about the data analyses and statistical tests used do not make it suitable for publication in the present form. In addition, the experimental design is not clearly described, and the methods lack important details and at times are a bit confusing, hindering the ability of this reviewer to fairly assess the findings. 

The language needs to be refined, some examples:

“Prolonged ingestion of ALAC, increases the ratio TRP/large neutral aminoacids (LNAA) in human plasma, which favors the entry of TRP into the rain.” The last word should be “brain”.

In the following sentence, the word “test” should be removed. 

“The effect of TRP and 5-HTP on motor coordination was assessed in test cohort 3, which included 34 wild type (WT) and Mecp2 Het female mice aged 16-20 weeks.” 

“The effect of THE VARIOUS treatments ON the rotarod (4-40 rpm in 300 s; Ugo Basile, Italy), was assessed in WT AND Mecp2 Het mice………………….”

The verb tense should be kept consistent:

“Regional analysis of monoamine levels showED a reduction of about 20% of 5-HT in the hippocampus, CEREBRAL cortex and brainstem, while no significant changes were observed in striatum. 5-HIAA levels were reduced by in the hippocampus and brainstem by 20% and 11% respectively, but no reduction was OBSERVED in the cortex AND striatum.”

The authors may want to properly write at least the first time that they are using the cerebral cortex. 

METHODS

these should be described with plenty of detail. The first paragraph is quite dense and a scheme describing the 4 cohorts would be helpful, perhaps they could add these to figure 1. The descriptions and details between paragraphs should be consistent.

Paragraph 2.1, the authors report: “The dose- and time-dependent effects of ALAC on brain and plasma levels of TRP and brain levels of indoles was assessed in 54 WT and 10 Mecp2 Het mice (Cohort 2).” 

End of paragraph 2.4: “Total levels of TRP in the whole brain, brain regions and plasma were determined as previously described [34].” However, there is no description of how the blood was collected from the animals and processed to obtain plasma.

The method used for the mice euthanasia should be reported.

Were the mice in cohort 3 and 4 only used for the behavioural assays or they were euthanised at the end of the treatment and behavioural tests to collect the brains (and the blood)?

It would be helpful if the authors would specify when the mice of each cohort were euthanised and for what measurements they were used. As suggested above the scheme in figure 1 could be developed to fully describe the experimental design with all the cohorts. 

At what time of the day were the animals euthanised and brains dissected? 

The sex of the animals in cohorts 1 and 2 should be reported. Were all the WT and mutant mice females? Was any male used?

How did the authors choose the animals age and the time of the injection of tryptophan?

Paragraph 2.2:

How were the doses of 5HT and Tryptophane chosen?

Missing information: 

Supplier for 5HT and ALAC 

How ALAC was dissolved/prepared and administered. 

“To establish the effect of ALAC on brain indoles, WT mice were treated with single doses of 250-8000 mg/kg orally. The dose of ALAC increasing brain TRP and indoles in WT mice (8000 mg/kg) was subsequently evaluated in Mecp2 Het mice to confirm its efficacy.” This should be in the results.

Paragraph 2.4:  please describe the composition of the lysis buffer. 

Data analyses and statistical tests: 

Why did the authors use different tests in the mice treated with ALAC? Did they compare the effects of ALAC on the 4 groups (WT +/- ALAC, Mecp2 Het +/-ALAC)? this should be done using at least a two-way ANOVA. The authors report: “One-way ANOVA followed by Dunnett’s test was used to compare the dose- and time-dependent effect of ALAC on TRP, 5-HT and 5-HIAA in WT mice. Indole levels in untreated WT and Mecp2 Het mice and in Mecp2 Het mice given ALAC or vehicle were compared by Student’s t-test.” This is confusing as the authors wrote that doses and times were tested on both genotypes: “The dose- and time-dependent effects of ALAC on brain and plasma levels of TRP and brain levels of indoles was assessed in 54 WT and 10 Mecp2 Het mice (Cohort 2).”  All comparisons should be well explained and likely the data should be reanalysed with the correct statistical tests.

The following sentence is confusing: “Brain (minus cerebellum) was divided into two hemispheres with a cut along the sagittal line. Half the brain was immediately frozen, while the cortex, hippocampus, striatum and brainstem were dissected free-hand from the remaining hemisphere.” 

Were the brain regions also frozen before the extracts were prepared? Were the half brains used in the Monoamines and Tryptophan Assays? Are these what the authors refer as “whole brain” in the results?

All the values generated by the statistical tests as well as the degrees of freedom should be reported, t(df)=X.XXXX; =x.xxxx ; F(DF,df)=X.XXX, =x.xxxx in the manuscript included the supplementary tables.

RESULTS

“Mecp2 Het mice aged 23-26 weeks at the time of sacrifice show reduced levels of brain 5-HT,….” The time of sacrifice should be reported since the levels of  5HT vary during the day and the night.

“The dose-dependent effect of ALAC on plasma and brain TRP and brain 5-HT and 5-HIAA in WT mice is shown in Figure 2” As mentioned in the methods there is no description of how the plasma was obtained.

Figure 3, these data are missing the controls, the effects of 8000 mg/kg ALAC in the Mecp2 het mice should be compared to the effects elicited by the same dose in WT mice. These data should be analysed using a Two-Way ANOVA with treatment and genotype as variables.

Figure 4 should be after figure Fig 2.

Paragraph 3.3

Please rewrite the sentence below to communicate that the effect were analysed in both genotypes or in Mecp2 Het in comparison to WT: “The effect of TRP and 5-HTP on the latency to fall from the accelerated rotarod was examined in Mecp2 Het mice at various time points after administration (Figure 5). As previously reported [4], the performance of untreated Mecp2 Het mice in the rotarod is much lower than in WT mice (mean latency to fall: Mecp2 Het, 159 ± 7 and WT, 273 ± 4 s).”

These data should be shown in the same graph for better comparison, also was genotype considered as a variable in the statistical analyses, the authors should use a three way ANOVA with genotype, time (days) and treatment as variables.

The supplementary data (suppl tables 1 and 2) are not cited or described in the manuscript. 

Why were the data shown in supplemental table 1 obtained using 11months old mice?  

LEGENDS

In the legends to the figures and tables all the abbreviations used should be explained at least the first time, including 5HT, 5HIAA etc.

The statistical test(s) should be reported in all of them.

Comments on the Quality of English Language

The language needs to be refined, some examples:

“Prolonged ingestion of ALAC, increases the ratio TRP/large neutral aminoacids (LNAA) in human plasma, which favors the entry of TRP into the rain.” The last word should be “brain”.

In the following sentence, the word “test” should be removed. 

“The effect of TRP and 5-HTP on motor coordination was assessed in test cohort 3, which included 34 wild type (WT) and Mecp2 Het female mice aged 16-20 weeks.” 

“The effect of THE VARIOUS treatments ON the rotarod (4-40 rpm in 300 s; Ugo Basile, Italy), was assessed in WT AND Mecp2 Het mice………………….”

The verb tense should be kept consistent:

“Regional analysis of monoamine levels showED a reduction of about 20% of 5-HT in the hippocampus, CEREBRAL cortex and brainstem, while no significant changes were observed in striatum. 5-HIAA levels were reduced by in the hippocampus and brainstem by 20% and 11% respectively, but no reduction was OBSERVED in the cortex AND striatum.”

The authors may want to properly write at least the first time that they are using the cerebral cortex. 

Reviewer 2 Report

Comments and Suggestions for Authors

The manuscript from Villani and colleagues with the title: “Boosting Serotonin synthesis is not sufficient to improve motor coordination of Mecp2 heterozygous mouse model of Rett syndrome addresses whether approaches to increase serotonin synthesis in the brain have an effect in rescuing the typical motor impairment observed in a mouse model for Rett syndrome. The authors first measure the plasma and brain levels of serotonin (5-HT), its main metabolite 5-HIAA and its amino acid precursor Tryptophan (TRP) in wildtype and Mecp2 heterozygous female mice. While serotonin is significantly reduced in the total brain and in different brain areas, except the striatum 5-HIAA is reduced only in the hippocampus and TRP not at all. Next the authors describe the effects on a treatment with α-Lactalbumin (ALAC) known to increase TRP levels. While ALAC increases TRP and 5-HIAA in the brain, the effect on 5-HT is very small. All the effects are limited to the first 2 hours after treatment. Finally, the authors assess whether a repeated treatment with TRP, 5-Hydroxytryptophan or ALAC may affect the motor impairment in Mecp2 heterozygous females. None of the treatments succeeds at rescuing the motor impairment in Mecp2 heterozygous mice or affects the motor performance on age- matched female WT mice.

This study is of interest, especially in the light of previous reports indicating the significant of the alterations in 5-HT levels in the brain of Mecp2 mutant mice as well as a therapeutic potential of promoting 5-HT signaling. However, there are a couple of points in the planning of the experiments that in my opinion weaken the conclusions presented by the authors and that I think should be addressed before the manuscript can be accepted for publication.

Below I list all my concern in a point to point manner:

Major concerns:

1)    While the measurements of the effects of a treatment with ALAC on 5-HT, TRP, 5-HIAA have been performed only for up to 8 hours after treatment, the analysis of the motor impairment has been performed at 1, 3, 7 and 14 days at a non-well specified time after administering a single injection only of the days of testing. Here I have a couple of concerns:

a)    The time between injection/oral administration and testing in the rotarod should be more clearly specified for each of the drugs in the description of the methods;

b)    For the drugs given orally/ in drinking water (ALAC and 5-HTP): were these given only of the day of treatment? Or over the 2 weeks of testing? The treatment schedule should be better described. Also does orally always means in the drinking water or rather in one administration as oral gavage? This is also not clear to me.

c)     The experiments do not provide a mean to know what happens to TRP, 5-HIAA and 5-HT after the repeated treatment. Is there any change in their levels over time? For drugs that have such a short half-life should one expect any cumulative effect over the days of treatment? Without this knowledge it is impossible to conclude that increasing 5-HT does not rescue the motor impairment in Mecp2 heterozygous mice, since no information is provided regarding the levels of 5-HT in the mice that are tested in the rotarod, or at least in mice that underwent the same treatment schedule.

d)    5-HTP was administered in the drinking water: please give more details on how the amount of 5-HTP taken by each mouse was controlled. Also, if I understand correctly 5-HTP was used only in the rotarod testing. So not data are provided showing its effect of TRP, 5-HT or 4-HIAA levels in the brain. These should be provided, at the time course of the treatment used for the testing of the motor performance.  

Taken together, I think either that the description of the experiments/results provided is not clear enough or that the data do not support the conclusion of the authors that increasing 5-HT in the Mecp2 heterozygous mice by different treatments does not rescue motor impairment. Indeed, I think that the demonstration that the different treatments increase in 5-HT in the Mecp2 heterozygous mice tested in the rotarod is not provided by the current data.

Minor points:

1)    Line 69: it is “brain” not “rain”

2)    Line 159: remove “by”

3)    In the discussion the authors say that the reduction in 5-HT in Mecp2 mutant mice may be underestimated since “the indole levels were normalized to tissue weight and the brain of Mecp2 mutant mice is smaller than the WT one”. I actually I am not sure what the authors mean here? Indeed normalizing per tissue weight is in my opinion the correct way to compare the levels in this scenario. Please explain this point more clearly

Comments on the Quality of English Language

only a couple of spelling mistakes

Round 2

Reviewer 1 Report

Comments and Suggestions for Authors

The authors should be commend for the thorough job in revising the paper.

The methods are well described, Figure 1 is very clear and helpful to understand the experimental design. The statistical test are appropriate and these findings will be a great contribution to the field.

Reviewer 2 Report

Comments and Suggestions for Authors

All my concerns have been addressed by the authors